# Tuning Wetting Properties Through Surface Geometry in the Cassie–Baxter State

**DOI:** 10.3390/biomimetics10010020

**Published:** 2025-01-02

**Authors:** Talya Scheff, Florence Acha, Nathalia Diaz Armas, Joey L. Mead, Jinde Zhang

**Affiliations:** Department of Plastics Engineering, University of Massachusetts Lowell, Lowell, MA 01854, USA; talya_scheff@student.uml.edu (T.S.); florence_acha@student.uml.edu (F.A.); nathalia_diazarmas@student.uml.edu (N.D.A.); joey_mead@uml.edu (J.L.M.)

**Keywords:** superhydrophobicity, contact angle hysteresis (CAH), Cassie–Baxter state, silicon micro-posts, surface geometry, dynamic wetting, photolithography

## Abstract

Superhydrophobic coatings are beneficial for applications like self-cleaning, anti-corrosion, and drag reduction. In this study, we investigated the impact of surface geometry on the static, dynamic, and sliding contact angles in the Cassie–Baxter state. We used fluoro-silane-treated silicon micro-post patterns fabricated via lithography as model surfaces. By varying the solid fraction (ϕ_s_), edge-to-edge spacing (L), and the shape and arrangement of the micro-posts, we examined how these geometric factors influence wetting behavior. Our results show that the solid fraction is the key factor affecting both dynamic and sliding angles, while changes in shape and arrangement had minimal impact. The Cassie–Baxter model accurately predicted receding angles but struggled to predict advancing angles. These insights can guide the development of coatings with enhanced superhydrophobic properties, tailored to achieve higher contact angles and customized for different environmental conditions.

## 1. Introduction

Superhydrophobicity is a unique surface phenomenon characterized by high water repellency, where surfaces exhibit a static water contact angle greater than 150° and a sliding angle less than 10° [1]. This property has significant applications in self-cleaning, anti-corrosion, drag reduction, anti-icing, anti-biofouling, anti-fogging, and water–oil separation [2]. One of the earliest recognized examples of a superhydrophobic surface is the lotus leaf, known for its high contact angle and self-cleaning properties. Initially, researchers attributed these properties solely to the high contact angle. However, the discovery of the rose petal, which also exhibits a high contact angle but retains water droplets, challenged this understanding.

To differentiate these behaviors, the concept of contact angle hysteresis (CAH) was introduced. CAH refers to the difference between advancing (θ_A_) and receding (θ_R_) angles, with the advancing angle being the maximum contact angle a droplet reaches before the three-phase contact line (TCL) moves, and the receding angle being the smallest angle when liquid is extracted from the droplet [3]. For example, while both the lotus leaf and the rose petal exhibit high advancing angles (>150°), their receding angles differ significantly: 150° for the lotus leaf and 104° for the rose petal [4,5]. As a result, the lotus leaf has low CAH, contributing to its self-cleaning property, while the rose petal’s high CAH causes water retention. This difference in CAH has guided the development of artificial superhydrophobic surfaces, mimicking the lotus leaf’s water-repellent properties.

In studies of wetting behavior, four key angles are typically considered: static, advancing, receding, and sliding angles. Static contact angles are measured when a droplet is placed on the surface and the needle is removed [3]. Advancing angles represent the maximum contact angle before the TCL moves. Receding angles indicate the smallest angle when liquid is removed from the droplet, and the sliding angle describes the angle of inclination at which the droplet begins to move due to gravity, often referred to as the “tilting angle” [3].

Theoretical models of surface wetting include Young’s Model, Wenzel’s Model, and Cassie–Baxter’s Model. Young’s Model describes a smooth surface with no roughness or chemical heterogeneity [6]. Wenzel’s Model accounts for roughness, where the liquid fully wets the textured surface [7]. In contrast, the Cassie–Baxter Model assumes that a liquid droplet is suspended on top of surface structures, trapping air beneath it and creating a rough, chemically heterogeneous interface [8]. The solid fraction (ϕ_s_), defined as the ratio of the solid–liquid contact area to the total area, and the edge-to-edge spacing (L) between surface structures are critical factors in determining whether the surface remains in the Cassie–Baxter state.

The Cassie–Baxter model is widely used to describe superhydrophobic surfaces due to its applicability to textured surfaces. However, its accuracy in predicting contact angles remains contested. Gao and McCarthy [9] challenged the model’s validity, showing significant discrepancies between predicted and experimental contact angles on heterogeneous surfaces. Others, including McHale [10], Panchagnula and Vedantam [11], and Nosonovsky [12], argued that the model is accurate when considering local areal fractions along the TCL. Extrand [13] further suggested that linear solid (referring to the proportions of solid and air gaps measured along the three-phase contact line) and air fractions along the TCL better predict contact angles than overall areal fractions.

Despite extensive research on static contact angles, dynamic angles—specifically advancing and receding angles—have been less explored in relation to the Cassie–Baxter model. These dynamic angles are crucial for practical applications of superhydrophobicity, as they determine a surface’s ability to repel or retain water. Additionally, the effect of surface geometry, such as solid fraction, edge-to-edge spacing, and micro-post shape and arrangement, on dynamic and sliding angles has not been thoroughly investigated [14].

In this study, we aim to bridge this gap by systematically investigating how variations in surface geometry affect wetting behavior in the Cassie–Baxter state. Using lithographically fabricated micro-post structures, we control solid fraction, edge-to-edge spacing, shape and arrangement of the posts. We hypothesize that the Cassie–Baxter model will accurately predict advancing and receding angles [9], that edge-to-edge spacing will have minimal impact on dynamic angles [15], and that changes in post shape and arrangement will increase advancing and receding angles [16,17]. Experimental data will be compared with the predictions of the Cassie–Baxter model to better understand how surface geometry influences wetting properties.

## 2. Experimental Section

### 2.1. Sample Design and Preparation

Three sets of micro-post structures were designed to evaluate the effects of surface geometry on wetting properties. Each set consisted of multiple samples fabricated on a 4-inch silicon wafer, with individual sections measuring 1.25 cm × 1.25 cm and spaced 0.75 cm apart. The design schematics for all samples were created using LayoutEditor™ 20241006 (Table 1).

In Set 1, the solid fraction (ϕ_s_) was varied to assess its impact on static, dynamic, and sliding contact angles. The solid fraction was calculated using the following equation:(1)$$\phi_{s}=\frac{\pi }{4{\left(\frac{L+d}{d}\right)}^{2}}$$
where d is the diameter of the micro-post and L is the edge-to-edge spacing between the posts.

In Set 2, the effect of edge-to-edge spacing on contact angles while keeping the solid fraction constant was explored. The edge-to-edge spacing L was calculated using:(2)$$L=d\left(\sqrt{\frac{\pi }{4\phi_{s}}}\;-1\right)$$

In Set 3, the shape and arrangement of the micro-posts were varied. This set included both square and circular posts, arranged in either square or hexagonal patterns. The solid fraction and edge-to-edge spacing were calculated using the following equations:(3)$$\phi_{s}=\frac{a^{2}}{L^{2}},\;\;\;L=\frac{a}{\sqrt{\phi_{s}}}$$
where L is the center-to-center spacing and a is the side length of square posts.

### 2.2. Fabrication of Silicon Micro-Post Structures

Figure 1 illustrates the fabrication process. A 4-inch silicon wafer (University Wafer) was cleaned with isopropanol and acetone, and dried with high-pressure nitrogen gas. The wafer was then loaded into the Brewer Science^®^ Cee^®^ 200CBX spin coater, and Shipley Microposit S1813 photoresist was applied, leaving a 2–4 mm border. Spin coating was carried out at 2000 rpm for 1 min. The coated wafer was soft-baked at 115 °C for 1 min to remove any residual solvent [18].

Once the wafer cooled, it was placed in the Heidelberg Direct Write uPG10 system for laser exposure. The exposure settings were set to 12 mW at 60% power in uni-directional mode. After laser writing, the wafer was transferred to the YES-58TA Image Reversal Oven for 90 min in an NH_3_ environment, rendering the exposed photoresist insoluble. A flood exposure was then performed using the Suss MicroTec MA6 Mask Aligner, and the exposure time was calculated based on the UV dose per second:(4)$$Exposure\;time=\;\frac{1600}{UV\;dose\;per\;sec}$$

The wafer was submerged in MicroChem Developer MF-26A for 40–60 s to remove the unexposed photoresist, followed by a DI water rinse and nitrogen drying.

The developed wafer underwent a hard bake at 115 °C for 1 min to strengthen the photoresist through cross-linking. Any excess photoresist was removed in an oxygen plasma strip (Oxford Plasmalab 80 Plus) for 1 min. To fabricate the micro-posts, the wafer was etched using the Oxford Plasmalab 100 ICP380 ICP Etch Tool, a process known as Deep Reactive-Ion Etching (DRIE). The parameters for etching 20 µm-tall micro-posts were set to a SF_6 flow rate of 40.0 sccm, O_2 flow rate of 5.0 sccm, and a forward power of 1000 W, with the etching process lasting 15 min [19].

After etching, the wafer was submerged in MicroChem Remover PG for 1–12 h to remove all remaining photoresist. The wafer was rinsed in DI water and dried using nitrogen gas. To ensure complete removal of any residual photoresist, the wafer was subjected to another round of oxygen plasma treatment for 1 min. For hydrophobicity enhancement, the wafer was treated with 20 µL of Sigma-Aldrich Trichloro Silane in a desiccator. A vacuum was applied for 2 min, followed by 20 min of vapor deposition. The wafer was then baked at 120 °C for 20 min to ensure uniform fluorination across the micro-post structures.

### 2.3. Measurements

Contact angle measurements were conducted using a Ramé-Hart Model 590 Advanced Automated Goniometer/Tensiometer (ramé-hart instrument co.), equipped with Automated Tilt, Dispensing, and DROPimage Advanced software. The methodology was adapted from Kwok et al. [20].

All experimental measurements were performed in a controlled laboratory environment maintained at room temperature (23 ± 2 °C) with a relative humidity of 45 ± 5% to ensure consistent testing conditions.

For the static contact angle, a 5 µL droplet of deionized (DI) water was dispensed onto the surface from a needle positioned 10 mm above. The needle was retracted to allow the droplet to remain on the surface. For superhydrophobic surfaces, a gentle shake of the needle was applied to release the droplet. The static contact angle was measured using the sessile drop method, with each measurement repeated three times at different locations, and the average value was calculated.

For the advancing angle measurement, a 10 µL droplet of DI water was dispensed from a needle positioned 15 mm above the surface, ensuring the needle was centered within the droplet. The advancing angle was determined by adding water at a flow rate of 0.25 µL/s until stabilization was achieved. Measurements were taken at three distinct locations, and the average angle was recorded (Figure 2).

To measure the receding angle, a 10 µL droplet was dispensed from a needle 15 mm above the surface. An additional 20 µL was dispensed to stabilize the droplet while ensuring the needle remained centered. The receding angle was measured by withdrawing water at a flow rate of 0.25 µL/s until the droplet began to de-pin from the surface, indicated by a significant change in droplet diameter. This process was repeated at three locations to obtain an average.

The sliding angle was measured by dispensing a 10 µL droplet, followed by an additional 20 µL to reach 30 µL while stabilizing the droplet. The needle was retracted to maintain droplet position on the surface, and the surface was tilted at a rate of 0.1°/s to minimize vibration. The angle at which the droplet began to de-pin was recorded, and measurements were taken at three locations to calculate the average sliding angle.

Figure 3 illustrates a representative tensiometer measurement of the receding contact angle. The image shows the moment when water is being withdrawn from the droplet, with the needle positioned centrally within the droplet. This precise measurement setup ensures accurate determination of the receding angle, which is critical for understanding the de-pinning behavior of water droplets from the surface.

Measurements on a smooth surface, considered ideal, with no imperfections, were conducted using the same methodology as for rough surfaces. The static contact angle was measured at three different locations, with the average calculated.

### 2.4. Governing Equations

The static, advancing, and receding angles from textured surfaces in Set 1 were plotted against the solid fraction. A separate graph comparing the experimental and accepted values of the advancing and receding angles using a modified version of the Cassie–Baxter equation:(5)$$cos\theta_{A}=\;r\phi_{s}\left(cos\theta_{Y}+1\right)-1\;\;\;$$
where cosθ_A_ is the apparent contact angle, r is the roughness factor, (ϕ_s_) is the solid fraction, and cosθ_Y_ is young’s contact angle. This allowed for the determination of accepted values (Figure 10) [8]. To validate the advancing and receding angle data, a graph of sinθ_sliding_ versus cosθ_R_ − cosθ_A_ was prepared. The results were compared to a linear fit derived from
(6)$$sin\alpha =\gamma \frac{Rk}{mg}\left(cos\theta_{R}-cos\theta_{A}\right)$$
where α is the sliding angle, γ is the surface tension, m is the mass of the droplet, and R, k are length scale and shape constants, respectively (with R typically being the droplet radius and k a fitting parameter based on experimental data) [4]. This equation is applicable only for small angles, indicating a critical threshold; for instance, a sliding angle of 21° appeared as an outlier on the graph (Figure 12).

Additionally, contact angle hysteresis (CAH) was calculated and graphed against the sliding angle as follows:(7)$$CAH=\theta_{A}-\theta_{R}\;\;$$
where θ_A_ is the advancing angle and θ_R_ is the receding angle.

In Set 2, a graph depicting the static, advancing, and receding angles was plotted against edge-to-edge spacing to examine any effects on angle measurements while maintaining a constant solid fraction of 0.05.

Set 3 involved generating a bar graph to compare the static, advancing, and receding angles across two distinct micro-post shapes and arrangements. This analysis aimed to identify measurable changes resulting from variations in shape or arrangement, ultimately determining which configuration yielded the highest contact angles.

## 3. Results and Discussion

### 3.1. Calculations and Design of Micro-Post Structures

The specifications that were varied are summarized in Table 1 and further detailed in the subsequent tables. The Set 1 sample consisting of circular micro-posts arranged in a square configuration. Set 2 maintained the same circular micro-post design and arrangement. Set 3 evaluated the effects of shape and arrangement, testing four distinct configurations: (a) square micro-posts in a square arrangement, (b) circular micro-posts in a square arrangement, (c) circular micro-posts in a hexagonal arrangement, and (d) square micro-posts in a hexagonal arrangement.

In Set 1 (Table 2), the solid fraction (ϕ_s_) was modified by adjusting the edge-to-edge spacing, while the micro-post diameter was held constant as defined in Equation (1). A lower solid fraction introduces more air between the pillars (resulting in a larger plastron), which maintains the water droplet in the Cassie–Baxter state, thereby enhancing hydrophobicity and increasing the receding angle [6]. However, a critical threshold exists, beyond which the spacing adversely affects the Cassie–Baxter state, allowing the weight of the droplet to surpass the surface tension (γ), thereby transitioning to the Wenzel state [21].

In Set 2 (Table 3), edge-to-edge spacing was varied by adjusting the diameter of the micro-posts while keeping the solid fraction (ϕ_s_) constant as described in Equation (2). An increase in spacing may facilitate the transition from the Cassie–Baxter to the Wenzel state.

In Set 3 (Table 4), two variables—shape and arrangement—were manipulated while maintaining a constant solid fraction (ϕ_s_). The shapes utilized in samples (a) and (b) were also employed in samples (c) and (d); however, the results obtained from the initial two samples elucidate whether changes in arrangement significantly influence static, dynamic, and receding contact angles.

### 3.2. Fabrication of Si Micro-Post Structures

The fabrication of Si micro-post structures necessitated various tests to ascertain the optimal parameters for accurate fabrication. The initial test involved determining the optimal exposure power and percentage during the direct writing step (c). Six samples were fabricated with varying exposure powers and percentages, as outlined in Table 5.

Sample 5, with an exposure power of 12 mW and a percentage of 60%, was identified as the optimal specification for direct writing, exhibiting clarity and well-defined edges. Other samples demonstrated non-circular shapes, which would compromise the circularity of the micro-posts and, consequently, their superhydrophobicity. During testing, uni-directional mode was disabled to reduce direct writing time; however, this decision resulted in a trade-off in the accuracy of micro-post shapes. In the experimental phase, the uni-directional mode was enabled to enhance the accuracy of the micro-post geometry (Figure 4).

Figure 4 demonstrates the improvement in micro-post geometry between the test and experimental phases. The left image shows the results from the test phase without the uni-directional mode. In contrast, the right image shows the experimental phase results with uni-directional mode enabled, displaying well-defined circular posts with sharp edges and consistent geometry. This improvement in post definition was crucial for achieving reliable and reproducible contact angle measurements.

When determining the optimal hard bake temperature, initial samples were subjected to a hard bake at 180 °C for 6 min. However, they exhibited reflow and Newton Rings (Figure 5). Reflow, characterized by thermal softening and rounding, occurred either due to the excessively high temperature, prolonged time, or a combination of both. Newton Rings arise from light reflections between spherical and flat surfaces. Following the challenges encountered at 180 °C for 6 min, a hard bake at 115 °C for 1 min was tested [22]. This modification resulted in significantly more accurate samples, thus establishing 115 °C for 1 min as the new hard bake recipe.

After optimizing the hard bake parameters, the optimal inductively coupled plasma (ICP) etch time necessary to achieve a height of 20 µm was determined. Three silicon wafer samples of the same design were prepared. Standard procedures were followed, varying the ICP etch duration—5 min, 10 min, and 15 min—while maintaining a constant O_2_ flow rate of 3.0 sccm. The heights of the etched pillars were subsequently measured using the Bruker DektekXT Profilometer, with the average heights plotted against etch time (Figure 6). The optimal etch duration was determined to be 15 min.

The final fabrication parameter assessed was the optimal O_2_ concentration during ICP etching. Five samples with varying O_2_ contents ranging from 0.0 sccm to 6.0 sccm were etched for 6 min each, aiming to identify the concentration that produced the straightest pillars. However, the initial tests yielded unfavorable results due to the micro-post diameters being excessively small, resulting in complete undercutting. In the subsequent trial, three samples with O_2_ concentrations ranging from 3.0 sccm to 5.5 sccm were etched for 15 min. This trial confirmed that an O_2_ concentration of 5.0 sccm produced pillars with highly accurate and straight sidewalls (Figure 7).

The optimal specifications for the fabrication of Si micro-posts are delineated in Table 6.

### 3.3. Contact Angle Measurements

The experimental static, advancing, and receding contact angles were plotted against solid fraction (ϕ_s_) as illustrated in Figure 8. A decrease in solid fraction correlates with an increase in both static and dynamic contact angles, as greater air volume is introduced between the micro-posts, enhancing the plastron effect. Notably, there was minimal variation observed between advancing angles, warranting further analysis (Figure 8). There is a critical threshold for solid fraction (ϕ_s_), beyond which additional increases in angles cease, a value to be elucidated in future research.

The theoretical values were computed and are shown in Figure 9. This equation necessitates values for θ_γ_, which represent the static and dynamic angles measured on a smooth surface, devoid of roughness (Table 7). When experimental data were compared to accepted theoretical curves, receding angles aligned closely with predictions; however, advancing angles deviated significantly. These findings suggest that while the Cassie–Baxter model provides an approximation for receding angles, a revised model is required to accurately predict advancing angles. The reason for the discrepancy in advancing angles remains unclear, but one hypothesis posits that advancing angles may approach 180° [7].

The experimental contact angle hysteresis (CAH) values were plotted against sliding angles, as demonstrated in Figure 10. The proportional relationship between CAH and sliding angle further corroborates the data from Set 1. Although no specific mathematical relationship exists between CAH and sliding angles, it is notable that CAH represents the difference between the advancing and receding angles, while the sliding angle reflects the difference between the cosines of advancing and receding angles.

The experimental sinθ_sliding_ was plotted against cosθ_R_ − cosθ_A_, as depicted in Figure 11. Strong correlation between the experimental and theoretical data illustrates the high accuracy of the advancing and receding angle data. Outlier results from the equation fitting were noted, which may arise from fitting angles only up to a certain degree. This confirms that (cosθ_R_ − cosθ_A_) can be utilized to predict the sliding angle.

The experimental static, advancing, and receding angles were graphed against edge-to-edge spacing (L) in Figure 12. As anticipated, no correlation was observed between changes in edge-to-edge spacing and static and dynamic contact angles. This observation corroborates trends seen in Set 1 (Figure 8), underscoring that solid fraction is a pivotal determinant influencing the differences between static and dynamic angles. Maintaining a constant solid fraction (ϕ_s_) in Set 2 did not yield discernible changes in contact angle measurements, further validating this hypothesis.

Similar trends were observed in sliding angle data for Set 2, as shown in Table 8.

Thus, without modifications to the solid fraction, no significant variation in angle measurements was detected, supported by a low standard deviation of 0.50.

Furthermore, the experimental static, advancing, and receding angles were evaluated against the shape and arrangement of micro-posts, as illustrated in Figure 13. Transitioning from circular to square shapes and from square to hexagonal arrangements yielded minimal changes in static and dynamic angles. The highest angles recorded were observed in square-shaped micro-posts arranged in both square and hexagonal patterns. The square shape exhibited the highest angles for static and advancing measurements, but it underperformed relative to circular shapes in receding angle measurements. Conversely, hexagonal arrangements with square shapes yielded the highest advancing and receding angles, although they lagged behind hexagonal/circular configurations in static angles. Future mathematical modeling will provide a deeper understanding of these data, elucidating the factors that contribute to elevated angles in specific configurations.

To evaluate the significance of alterations in shape and arrangement, sliding angle data were scrutinized (Table 9). There were instances where Set 3 data exceeded those of the other sets; however, such discrepancies can be attributed to variances in specifications. With a solid fraction of 0.05, the sliding angles consistently remained lower than those in Sets 1 and 2 with a solid fraction (ϕ_s_) of 0.05. The consistently low sliding angles indicate the potential for further reductions when solid fraction (ϕ_s_) is minimized.

## 4. Conclusions

This study provides valuable insights into the influence of surface geometry on wetting behavior in the Cassie–Baxter state. By systematically varying factors such as solid fraction, edge-to-edge spacing, and micro-post shape and arrangement, we were able to elucidate their respective impacts on static, advancing, receding, and sliding contact angles. The results demonstrate that solid fraction is the key determinant, with a decreasing solid fraction leading to increased hydrophobicity and higher contact angles. In contrast, changes in edge-to-edge spacing and micro-post shape/arrangement had relatively minor effects. The experimental data were also compared to theoretical models, revealing that the Cassie–Baxter equation accurately predicted receding angles but struggled with advancing angles, suggesting the need for refinements to the model.

This study focused on silicon substrates with fixed mechanical properties. Recent literature [23] suggests that the substrates’ elasticity could offer additional opportunities to optimize superhydrophobicity performance. For instance, elastic substrates may allow for dynamic modification of the surface geometry while the substrate is under load, which can potentially lead to the tuning of the wetting behavior.

Overall, these findings provide important design guidelines for engineering superhydrophobic surfaces with tailored wetting properties for various applications, such as self-cleaning, anti-corrosion, and drag reduction. Future work should explore the limits of the Cassie–Baxter state and investigate additional geometric factors that may influence dynamic wetting behavior.

## Figures and Tables

**Figure 1 biomimetics-10-00020-f001:**
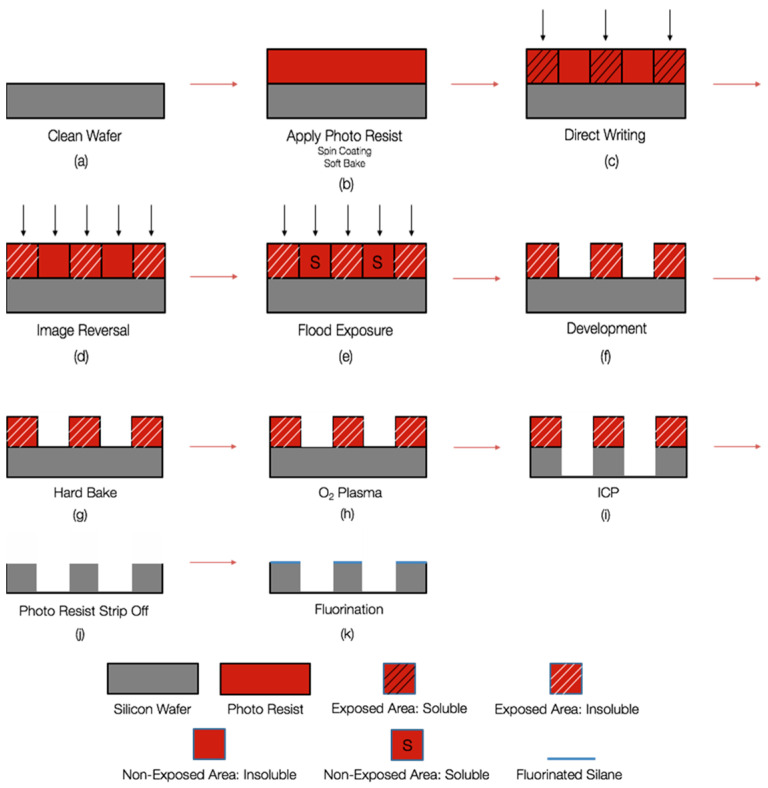
Fabrication processes for all silicon wafers.

**Figure 2 biomimetics-10-00020-f002:**
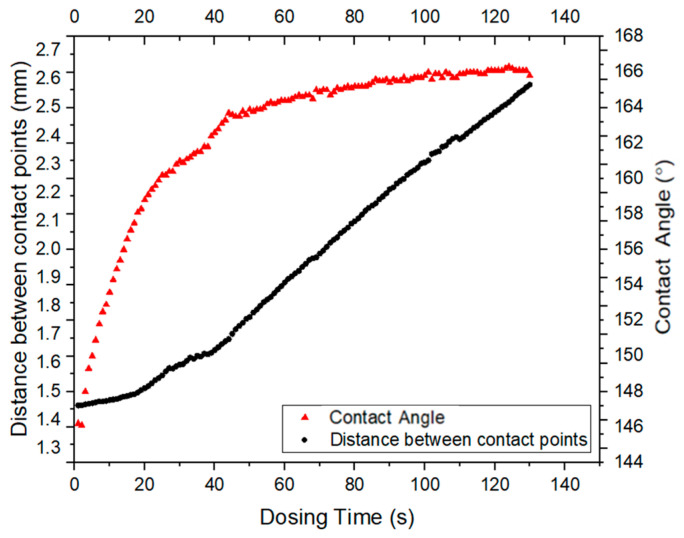
Tensiometer reading of advancing contact angle.

**Figure 3 biomimetics-10-00020-f003:**
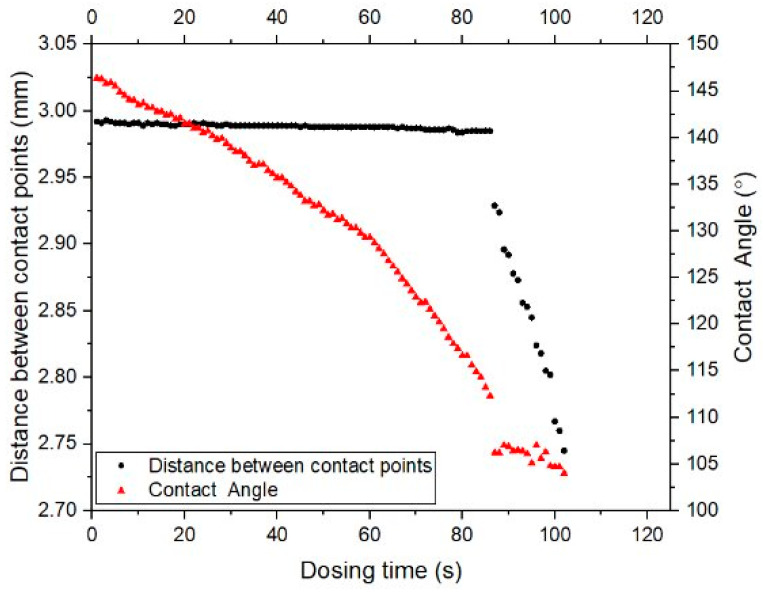
Tensiometer reading of receding contact angle.

**Figure 4 biomimetics-10-00020-f004:**
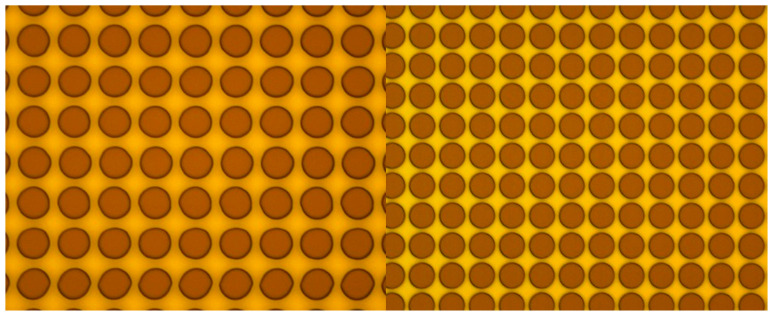
(**Left**) Test Phase and (**Right**) Experimental Phase.

**Figure 5 biomimetics-10-00020-f005:**
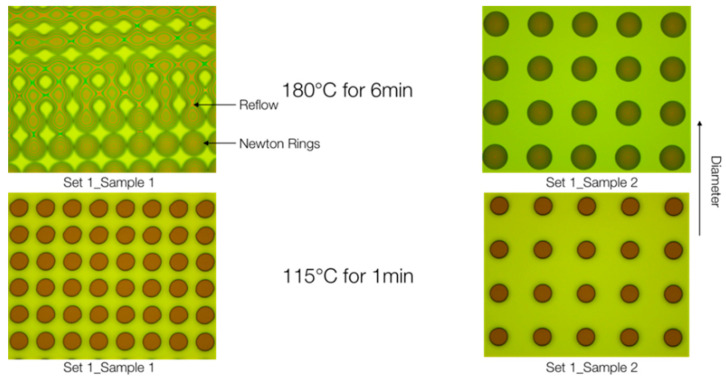
Samples with varied hard bake temperature and times.

**Figure 6 biomimetics-10-00020-f006:**
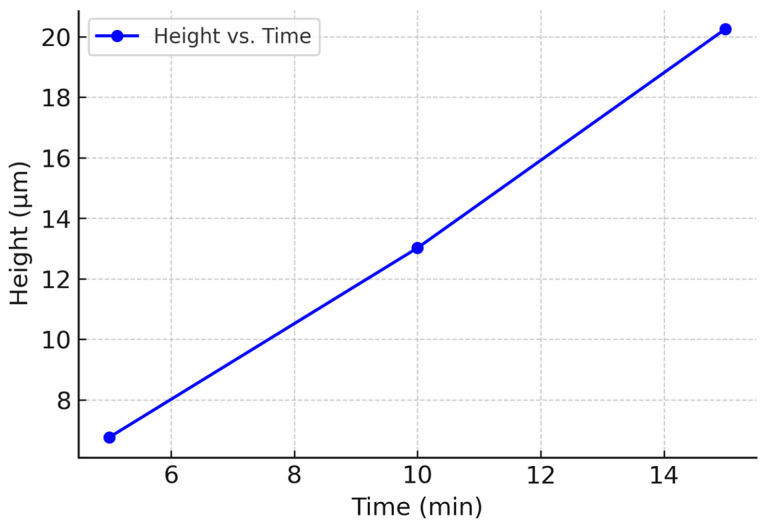
Graph used to determine optimal ICP etch time.

**Figure 7 biomimetics-10-00020-f007:**
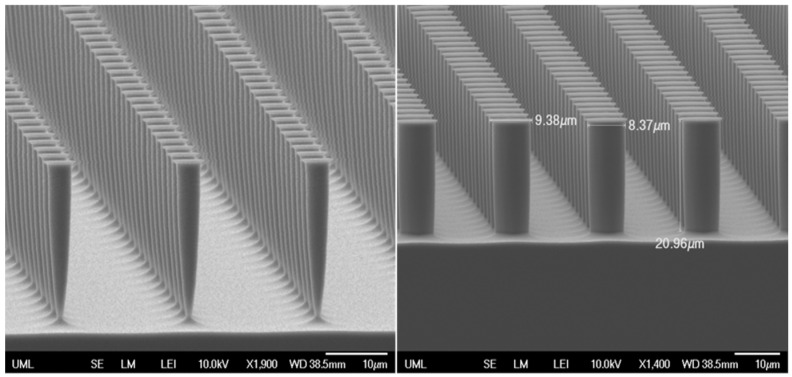
(**Left**) Undercut sample and (**Right**) optimal 5.0 sccm sample.

**Figure 8 biomimetics-10-00020-f008:**
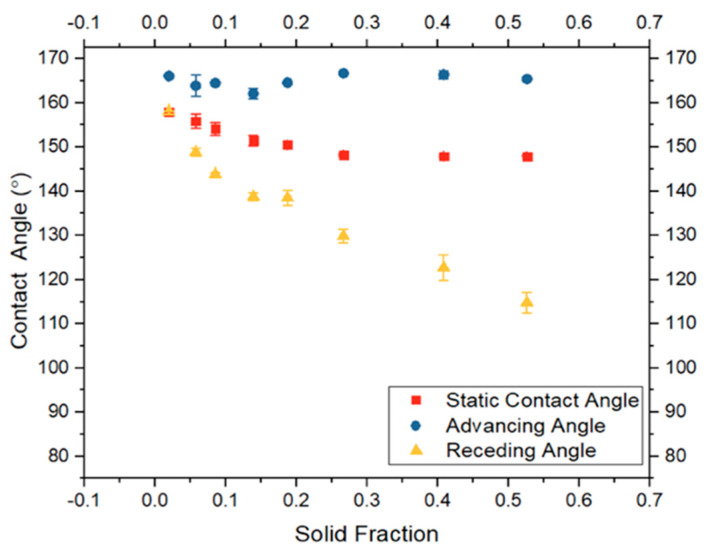
The effect of solid fraction on static and dynamic contact angle.

**Figure 9 biomimetics-10-00020-f009:**
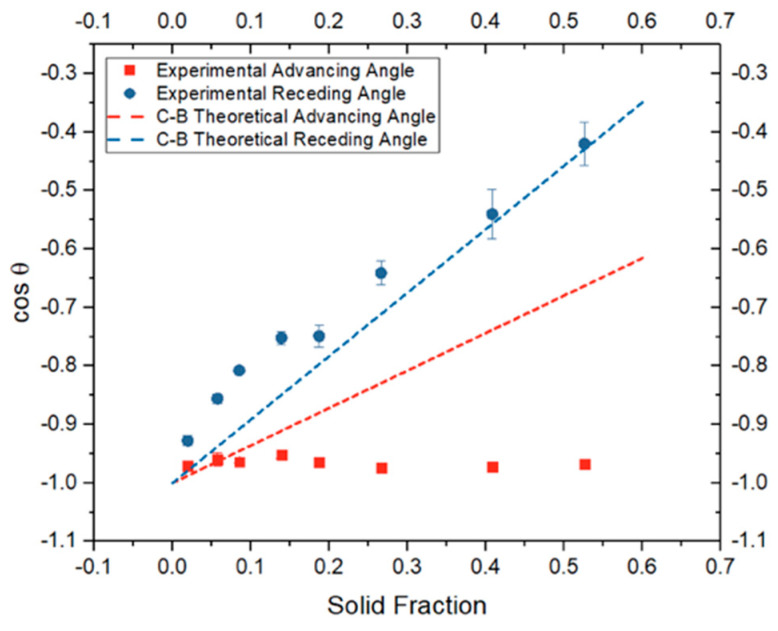
Comparing experimental dynamic angles to theoretical values predicted by Cassie–Baxter equation.

**Figure 10 biomimetics-10-00020-f010:**
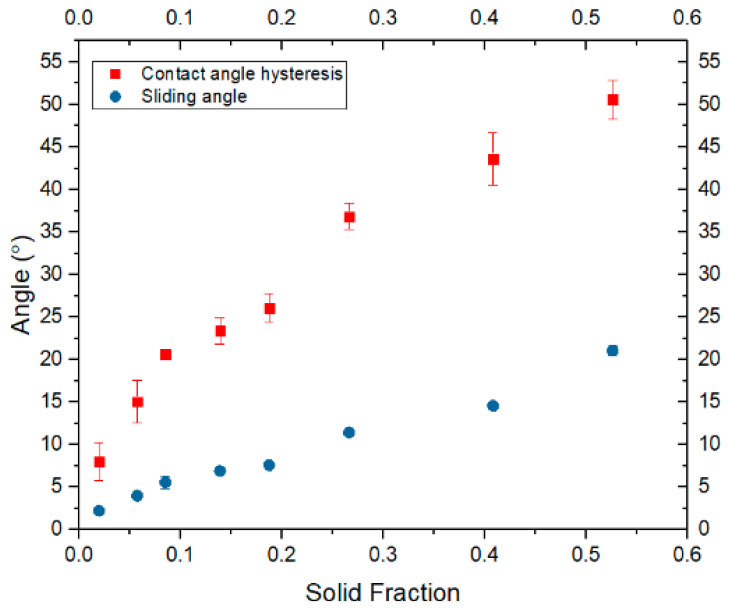
Contact angle hysteresis (CAH) compared to sliding angle.

**Figure 11 biomimetics-10-00020-f011:**
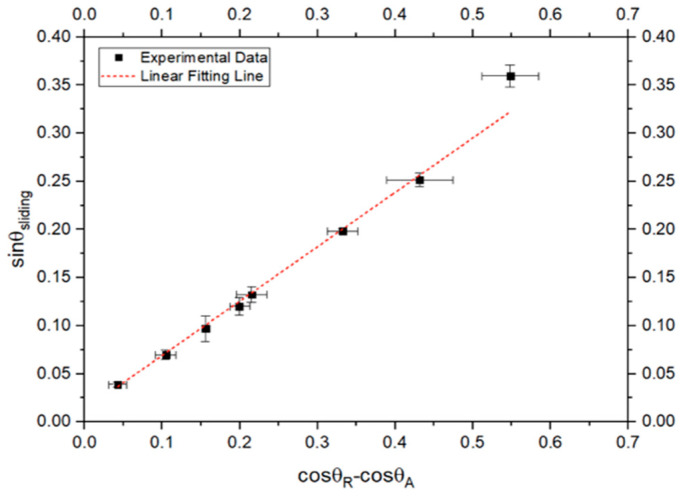
Comparing the sin of the sliding angle with the cos of the receding angles minus the cos of the advancing angle.

**Figure 12 biomimetics-10-00020-f012:**
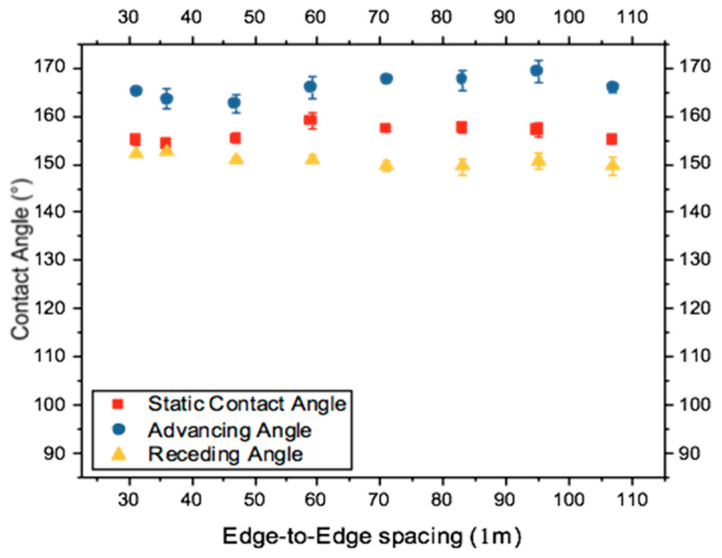
Comparing static and dynamic angles when edge-to-edge spacing is changed.

**Figure 13 biomimetics-10-00020-f013:**
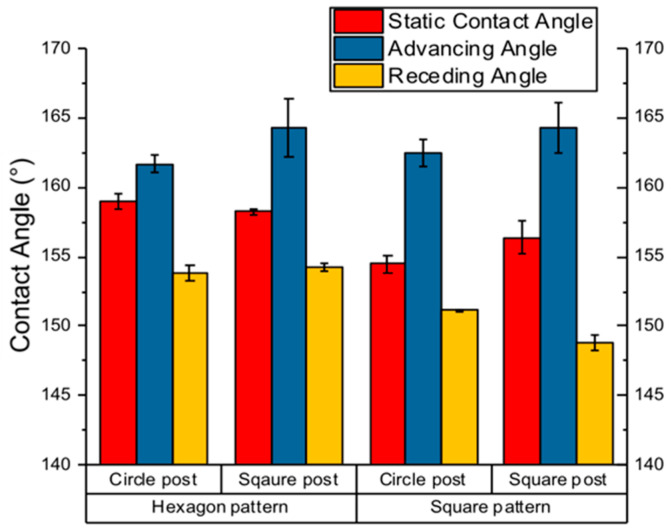
Comparing static and dynamic angles when shape and arrangement is changed.

**Table 1 biomimetics-10-00020-t001:** Dimensions of the Surfaces are cited.

Shape and Arrangement	Micro-Post Height (µm)	Micro-Post Diameter (µm)	Edge-to-Edge Spacing (µm)	Solid Fraction
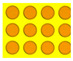	20	16	2.5–80	21–58
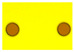	20	10–36	30–107	5%
a. 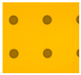	20	16	a.42	5%
b. 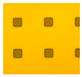			b.47	
c. 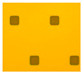			c.67	
d. 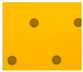			d.75	

**Table 2 biomimetics-10-00020-t002:** Parameters that varied for Set 1.

Sample	1	2	3	4	5	6	7	8
Edge-to-Edge Spacing (µm)	2.5	5	10	15	20	30	40	80
$\mathrm{Solid}\;\mathrm{fraction}\;\left(\phi_{s}\right)$	59%	46%	30%	21%	16%	10%	6.4%	2.2%

**Table 3 biomimetics-10-00020-t003:** Parameters that varied for Set 2.

Sample	1	2	3	4	5	6	7	8
Diameter (µm)	10	12	16	20	24	28	32	36
Edge-to-Edge Spacing (µm)	30	36	47	59	71	83	95	107

**Table 4 biomimetics-10-00020-t004:** Parameters that varied for Set 3.

Sample Letter	(a)	(b)	(c)	(d)
Edge-to-Edge Spacing (µm)	42	47	—	—
Center-to-Center Spacing (µm)	—	—	67	75

**Table 5 biomimetics-10-00020-t005:** Exposure Power and Percentages.

Sample	Exposure Power (mW)	Percentage (%)
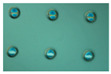	6	60
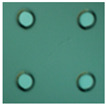	6	90
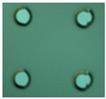	9	60
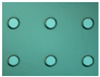	9	90
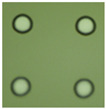	12	60
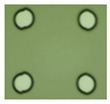	12	90

**Table 6 biomimetics-10-00020-t006:** Machine Settings for Fabrication.

Exposure Power (mW)	12 mW
Percentage (%)	60%
Soft Bake Temperature (°C) and Time (min)	115 °C/1 min
Hard Bake Temperature (°C) and Time (min)	115 °C/1 min
ICP Etch Time (min)	15 min
$O_{2}$ Concentration (sccm)	5.0 sccm

**Table 7 biomimetics-10-00020-t007:** Angles on smooth surfaces (θ_γ_).

Types of Angles	Static Contact Angle (°)	Advancing Angle (°)	Receding Angle (°)
Smooth Surface	110.1	123.4	94.6

**Table 8 biomimetics-10-00020-t008:** Sliding angle data when edge-to-edge spacing is changed.

Sample	1	2	3	4	5	6	7	8
Sliding Angle (°)	2.8	3.2	3.4	4.2	4.1	3.6	4.0	4.0

**Table 9 biomimetics-10-00020-t009:** Sliding angle when shape and arrangement are changed.

Sample Letter	(a)	(b)	(c)	(d)
Sliding Angle (°)	3.8	3.2	2.5	2.6

## Data Availability

Data are unavailable due to privacy or ethical restrictions.

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
