# Peer review of "Tuning Wetting Properties Through Surface Geometry in the Cassie–Baxter State"

_biomimetics, 2025, doi:10.3390/biomimetics10010020_

Round 1

Reviewer 1 Report

Comments and Suggestions for Authors

The authors investigated the impact of surface geometry on the static, dynamic, and sliding contact angle in the Cassie-Baxter state. This research is very important for understanding the tuning wetting properties by adjusting surface geometry. Generally, the experiments are thoroughly investigated, and the experimental results are good. Thus, I recommend its publication in Biomimetics after answering the following questions:

(1) In Figures, the terms ‘Receding angle’ and ‘Advancing angle’ are suggested to be revised as ‘Receding contact angle’ and ‘Advancing contact angle’.

(2) The authors said in the text, ‘The Cassie-Baxter model accurately predicted receding angles but struggled to predict advancing angles’ and ‘These insights in this work can guide the development of coatings with enhanced superhydrophobic properties’. It can be found in some literature (Langmuir, 2022, 38, 18-35) that the enhanced superhydrophobic properties not only correlates with surface geometry, but also is related with the Young’s modulus of substrates. I suggest that the authors can discuss the enhanced superhydrophobic property of elastic substrates in the Conclusion Part, which may be related with the experimental results.

Have a good luck.

Reviewer 2 Report

Comments and Suggestions for Authors

The quality of the presentation should be improved. 

(1) The humidity and temperature during contact angle measurements should be reported, since even lotus leaf is not super-hydrophobic (see, Cheng and Rodak, https://doi.org/10.1063/1.1895487) when water is condensed on the lotus leaf.  

(2) "Linear solid" should be defined.

(3) Figures 3 and 4 are not discussed in the text. 

(4) "Derivation of Equations" is inaccurate since the authors did not derive the equations, but simply stated them without providing references. 
